# Sarcopenia Is Associated with Metabolic Syndrome in Korean Adults Aged over 50 Years: A Cross-Sectional Study

**DOI:** 10.3390/ijerph19031330

**Published:** 2022-01-25

**Authors:** Do-Youn Lee, Sunghoon Shin

**Affiliations:** 1Research Institute of Human Ecology, Yeungnam University, Gyeongsan-si 38541, Gyeongbuk, Korea; triptoyoun@yu.ac.kr; 2Neuromuscular Control Laboratory, Yeungnam University, Gyeongsan-si 38541, Gyeongbuk, Korea

**Keywords:** sarcopenia, metabolic syndrome, Korean

## Abstract

This study assessed the association between sarcopenia and metabolic syndrome in Korean adults aged over 50 years. The study obtained data from the Korea National Health and Nutrition Examination Survey (KNHANES, 2008–2011), a cross-sectional and nationally representative survey conducted by the Korean Centers for Disease Control and Prevention. Among the 8363 participants included in this study, the prevalence rate of sarcopenia according to metabolic syndrome was stratified by sex. Crude odds ratios not adjusted for any variables were 1.827 (1.496–2.231) in males, 2.189 (1.818–2.635) in females, and 2.209 (1.766–2.331) in total participants compared with non-sarcopenia. Model 3, which was adjusted for all variables that could affect sarcopenia and metabolic syndrome, showed significant increases in the odds ratios, to 1.957 (1.587–2.413) in males, 1.779 (1.478–2.141) in females, and 1.822 (1.586–2.095) for total participants. The results suggest that the association between sarcopenia and metabolic syndrome is significant in Korean adults.

## 1. Introduction

Sarcopenia is described as a loss of muscle mass and strength because of changes in body composition [1]. The mechanism of sarcopenia remains unknown; however, it is related to various conditions such as a lack of activity, malnutrition, aging, and changes in levels of hormones, including cortisol and testosterone [2]. Furthermore, the increased body fat associated with sarcopenia patients’ decreased muscle mass is linked to cardiovascular disease, diabetes, physical impairment, and mortality [3,4].

In Korea, the prevalence of sarcopenia among the elderly, aged 70 or older, was 18.4% in 2017 [5]. In addition, sarcopenia affected more than 50 million adults globally in 2000, and this number is expected to increase to more than 200 million by 2040 [6]. The socioeconomic cost of sarcopenia in the United States was estimated to be $18.5 billion in 2000, and, thus, additional studies on sarcopenia are required [7].

Metabolic syndrome (MetS) is a collection of metabolic disorders (abdominal obesity, hypertension, high blood glucose level, and abnormal blood lipids) related to an elevated risk of cardiovascular morbidity and death, as well as all-cause mortality [8]. Insulin resistance is the most common cause of MetS [9]. As blood glucose levels increase in response to increased insulin resistance, insulin secretion increases further, resulting in hyperinsulinemia, which limits sodium excretion in the kidneys and causes hypertension [10]. It also decreases high-density lipoprotein cholesterol (HDL-C) levels and increases triglyceride (TG) levels, resulting in dyslipidemia [11].

Sarcopenia is a geriatric and aging illness that has been linked to various metabolic diseases, such as obesity, insulin resistance, diabetes, dyslipidemia, and hypertension [12,13,14]. Furthermore, the increased body fat associated with metabolic disorders has been found in people with sarcopenia, owing to reduced muscle mass [3]. An association between sarcopenia and MetS has been suggested by several studies; however, this association remains unclear. Various studies have identified a link between sarcopenia and MetS, while other studies have shown that the relationship is dependent on gender [15,16,17]. However, these studies were only conducted for subjects who were too old or suffered from certain other diseases. Moreover, information on the link between sarcopenia and MetS from population-based studies in Korea is insufficient. In addition, the change in skeletal muscle mass, strength, and power of people aged 30 to 50 is not significant, while the change is noticeable from the age of 50 or older [18,19]. Consequently, we investigated the association between sarcopenia and MetS, by focusing on Korean adults aged ≥50 years, owing to the significant loss of muscle mass in this age group.

## 2. Materials and Methods

### 2.1. Data Source and Sampling

This study used data from the Korea National Health and Nutrition Survey (KNHANES, 2008–2011) conducted by the Korean Centers for Disease Control and Prevention. The participants responded to both an examination and health survey of adults aged 50 years or older. Among 37,753 participants who participated in the KNHANES, there were 23,671 participants aged <50 years, 1520 participants for whom MetS components were not assessed, and 3666 participants for whom sarcopenia variables were not assessed, while 539 non-participants in the health survey were excluded. Consequently, 8363 participants were included in this study (Figure 1).

### 2.2. Measurement of Variables

#### 2.2.1. Covariates

BMI was calculated by dividing weight (kg)/height (m^2^). Smoking status was categorized as never smokers, ex-smokers, and current smokers. Drinking condition was dichotomized into current users and non-users. Marital status was classified as living with a spouse or without a spouse. Individual income was divided into quartiles. Physical examinations included height, weight, fasting glucose, waist circumference, triglyceride, diastolic and systolic blood pressure, HDL-cholesterol (HDL-C), and total cholesterol measurement variables. Blood samples were collected from participants in the morning after overnight fasting and were analyzed at a national central laboratory. Blood pressure was measured using a mercury sphygmomanometer, with participants in a seated position after a 10 min rest period. Two measurements were made for all participants at 5 min intervals. An average of the two measurements was used for data analyses.

#### 2.2.2. Sarcopenia

Appendicular skeletal muscle mass (ASM) was measured using dual X-ray absorptiometry (QDR4500A; Hologic, Inc., Bedford, MA, USA). The sarcopenia muscle mass index (SMI) was calculated as ASM (kg)/body mass index (BMI, kg/m^2^). Sarcopenia was defined as an SMI <0.789 in males and <0.521 in females, based on the criteria of the Sarcopenia Project [6].

#### 2.2.3. Metabolic Syndrome

MetS in the study was diagnosed if three or more of the five components were satisfied using the guidelines of the National Cholesterol Education Program Adult Treatment Panel III (NCEP-ATP III): (1) abdominal obesity: waist circumference >90 cm in men and >85 cm in women; (2) hypertriglyceridemia: ≥150 mg/dL; (3) reduced HDL-C: <40 mg/dL for men and <50 mg/dL for women; (4) hypertension: systolic blood pressure ≥130 mmHg or diastolic blood pressure ≥85 mmHg; and (5) elevated fasting glucose: ≥100 mg/dL. If participants were using anti-hypertension, diabetes, or dyslipidemia treatment medication, they were present [20].

### 2.3. Data Analysis

Since this study uses complex sample data, the weighting given by the KNHANES has been applied. Data were expressed as absolute numbers and estimated percentages (with standard errors) or as mean ± standard deviation (SD). The survey responses were weighted by reference to a multistage, complex probability sampling design. General characteristics were compared according to sarcopenia and MetS using the chi-square test. Multivariate logistic regression analysis was used to analyze the association between sarcopenia and spirometry patterns and *p*-values <0.05 were considered statistically significant. Data analysis was performed using SPSS 27.0 Window’s version.

## 3. Results

### 3.1. Characteristics of Participants According to Sarcopenia and Sex

Both males and females showed a significantly higher prevalence of MetS in participants with sarcopenia (Sarcopenia/Non-sarcopenia: 49.6/35.0 in males, 55.1/35.9 in females). In men, all variables showed significance, except for individual income, diastolic blood pressure, and total cholesterol. In contrast, significance was shown in all variables, except smoking status, individual income, weight, and diastolic blood pressure, in females (Table 1).

### 3.2. Characteristics of Participants According to MetS and Sex

In this study, the prevalence of sarcopenia according to MetS differed in both males and females (MetS/Non-MetS: 22.0/13.4 in males, 27.3/14.7 in females). Age, smoking and drinking status, marital status, individual income, and total cholesterol did not show significance in males. However, only smoking status did not show significance in females (Table 2).

### 3.3. MetS Components According to Sarcopenia and Sex

The five components of MetS, namely high fasting glucose level, abdominal obesity, high TG level, high blood pressure, and low HDL-C level, were all higher in males with sarcopenia. However, there were no significant differences in HDL-C levels in female (Table 3).

### 3.4. Odds Ratios for Sarcopenia According to MetS Stratified by Sex

The prevalence of sarcopenia according to MetS was classified according to sex. Crude odds ratios that were not adjusted for any variables showed that the probability of sarcopenia was 1.827 (1.496–2.231) in males, 2.189 (1.818–2.635) in females, and 2.209 (1.766–2.331) for the total participants, compared with non-sarcopenia. In model 3, which was adjusted for variables that could affect sarcopenia and MetS, significant increases in the probabilities were observed: 1.957 (1.587–2.413) in males, 1.779 (1.478–2.141) in females, and 1.822 (1.586–2.095) for total participants (Table 4).

## 4. Discussion

In this study of Korean adults aged over 50 years, the main findings are that sarcopenia is independently related to MetS, after adjusting for various variables including age, sex, smoking and drinking status, and individual income.

Both males and females had a significantly higher average age in people with sarcopenia. This result can be interpreted to mean that muscle mass decreases with age. This is consistent with the findings of a previous study, namely a gradual decrease in muscle mass was noted after reaching the maximum muscle mass in people in their 30’s, a 1–2% loss of muscle mass is noted every year in those in their 50’s, and the muscle mass is reduced by half in those in their 70’s [21].

In Table 2, the characteristics of participants according to the prevalence of MetS showed significant differences between males and females in the prevalence of sarcopenia. Table 3 shows significant differences in all variables in males, and significant differences in all variables except HDL-C in females in the analysis of MetS components following sarcopenia. The relationship analysis of sarcopenia with MetS in Table 4 shows that the association is also high at 1.822 (1.586–2.095) in model 3, which adjusted the variables that could affect it. Therefore, a correlation between the two was evident.

A previous study revealed that the association between sarcopenia and MetS was only found in men, which was different from the results of this study [17]. This difference is thought to be due to the age of the study subjects. In this previous study, the average age of the study subjects was 73.1 years in males and 72.8 years in females, and the proportion of the elderly in their 70’s or older was high. In addition, in the previous study, the age group of the subjects was high, regardless of sarcopenia, so there was no significant difference between groups in the MetS factor excluding WC [17]. For this reason, it is thought that there may have been a difference from the results of this study. Another study also revealed that there was only association between sarcopenia and metabolic syndrome in males, but it is believed that there was a difference from our study, as the subjects of this study were limited to chronic obstructive pulmonary disease patients [15].

Although the pathological mechanisms underlying the relationship between sarcopenia and MetS remain unknown, there may be several shared causes. First, the five components of MetS are abdominal obesity, high blood pressure, high blood glucose, high TG, and low HDL-C [22]. Insulin resistance and inflammation have been hypothesized to be the primary factors underlying MetS [20]. Under normal conditions, skeletal muscle is essential for systemic glucose homeostasis, accounting for 80% of normal glucose absorption and metabolism via insulin stimulation [23]. Second, IL-6, IL-8, IL-15, fibroblast growth factor 21, irisin, myonectin, and myostatin are among the myokines produced and secreted by skeletal muscle cells [24]. Exercise and muscle exertion control the majority of myokines. They have positive effects on glucose and lipid metabolism, as well as inflammation, by counteracting the negative effects of inflammatory cytokines [25]. Third, by lowering the basal metabolic rate, a reduction in muscle mass leads to an increase in fat mass [26]. Increased fat levels cause inflammatory cytokines, such as tumor necrosis factor-alpha and interleukin (IL)-634, to be secreted, as well as insulin resistance, which contribute to MetS [27,28]. Thus, a decrease in muscle mass results in an increase in insulin resistance, which leads to MetS and type 2 diabetes, as it is the major location for glucose and fatty acid metabolism. The metabolic effect of adipokines secreted by adipose tissue has been shown to reverse the pathological consequences of metabolic disorders by secreting proteins or myokines from skeletal muscle fibers (especially type IIb muscle fibers), and these muscle fibers are selectively lost during the aging process [29].

Despite several significant findings, this study had some limitations. First, it was a cross-sectional study that assessed both sarcopenia and MetS variables simultaneously; thus, while this study may provide additional insights into the nature of this association, it was difficult to determine the order of the underlying causes of sarcopenia, because a cross-sectional study design tends to cause confusion regarding the temporal sequence. Therefore, identifying a method to elucidate the causal relationship between sarcopenia and MetS through longitudinal research will be important in the future. Second, because there were no KNHANES data on types and dosage of medication, medicines that may impact sarcopenia and MetS could not be ruled out. Third, there was a draw-recall bias because the socio-statistical characteristics of the research population were gathered through questionnaires. Finally, because the information on the components of MetS was inadequate, the total prevalence of MetS may have been underestimated. However, this process was most likely randomly eliminated and is unlikely to have had a major influence on the study results. Despite these limitations, the scholarly and clinical implications of this study are substantial. The strength of this study is that it collected data from a large Korean population with a high response rate, allowing for numerous statistical modifications to account for potential causes of disruption.

## 5. Conclusions

To summarize the study results, after adjusting for different confounding variables, such as age, sex, smoking and drinking behavior, and individual income, the primary finding of this study of Korean adults aged over 50 years was that sarcopenia is independently related to MetS.

## Figures and Tables

**Figure 1 ijerph-19-01330-f001:**
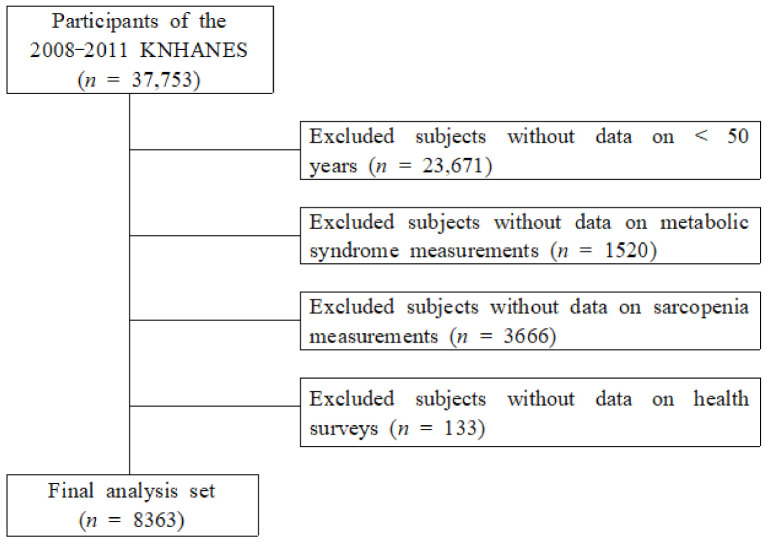
Selection of participants from the Korea National Health and Nutrition Examination Survey 2008–2011.

**Table 1 ijerph-19-01330-t001:** Participant characteristics according to sarcopenia and sex.

Variables	Males			Females		
	Sarcopenia (*n* = 681)	Non-Sarcopenia (*n* = 2946)	*p*	Sarcopenia (*n* = 951)	Non-Sarcopenia (*n* = 3785)	*p*
Age (y)	64.75 ± 0.41	59.93 ± 0.19	<0.0001	66.10 ± 0.42	61.04 ± 0.19	<0.0001
MetS, *n* (%)	331 (49.6)	978 (35.0)	<0.0001	527 (55.1)	1372 (35.9)	<0.0001
BMI (kg/m^2^), mean (SD)	25.01 ± 0.13	23.61 ± 0.07	<0.0001	26.07 ± 0.16	23.81 ± 0.06	<0.0001
<18.5 (underweight), *n* (%)	8 (1.2)	120 (3.4)		3 (0.7)	118 (2.9)	
<25 (normal-weight), *n* (%)	346 (48.3)	1982 (65.4)		375 (40.8)	2454 (64.7)	
≥25 (overweight), *n* (%)	327 (50.5)	844 (1.0)		573 (58.5)	1213 (32.4)	
Smoking status, (%) (current-/ex-/non-smoker)	56.4/31.1/12.4	60.3/23.4/16.2	0.002	6.5/2.7/90.9	7.2/1.6/91.2	0.200
Drinking status (%) (current-/non-drinking)	64.5/35.5	71.6/28.4	0.002	22.9/77.1	28.4/71.6	0.005
Marital status, (%) (living with spouse)	88.3	92.7	0.003	55.0	70.3	<0.0001
Income (individual)						
Q1 (lowest) Q2 Q3 Q4 (highest)	25.7 27.2 26.3 20.8	24.0 25.5 24.0 26.5	0.121	26.5 23.6 27.9 21.9	24.4 25.2 25.5 24.9	0.221
Height (cm)	160.85 ± 0.25	168.35 ± 0.12	<0.0001	148.14 ± 0.22	155.07 ± 0.12	<0.0001
Weight (kg)	64.91 ± 0.43	67.05 ± 0.24	<0.0001	57.36 ± 0.41	57.36 ± 0.17	0.996
Fasting glucose (mg/dL)	109.79 ± 1.48	104.53 ± 0.55	0.001	103.81 ± 1.03	100.49 ± 0.45	0.003
Waist circumference (cm)	87.98 ± 0.37	84.95 ± 0.20	<0.0001	86.42 ± 0.42	81.43 ± 0.21	<0.0001
Triglyceride	183.29 ± 6.95	160.78 ± 3.33	0.005	148.50 ± 3.14	133.89 ± 1.78	<0.0001
Systolic BP (mmHg)	130.44 ± 0.87	126.27 ± 0.44	<0.0001	131.30 ± 0.74	126.06 ± 0.42	<0.0001
Diastolic BP (mmHg)	79.90 ± 0.57	80.78 ± 0.29	0.143	77.94 ± 0.39	77.83 ± 0.23	0.802
HDL-C	43.82 ± 0.53	45.42 ± 0.28	0.005	47.94 ± 0.46	48.70 ± 0.26	<0.0001
Total cholesterol	185.81 ± 1.97	187.59 ± 0.79	0.400	205.42 ± 1.36	199.76 ± 0.78	<0.0001
Skeletal muscle index	0.74 ± 0.00	0.91 ± 0.00	<0.0001	0.48 ± 0.00	0.61 ± 0.00	<0.0001

Data are presented as means ± SD or number (%). MetS; metabolic syndrome, BMI; body mass index, BP; blood pressure HDL-C; high density lipoprotein-cholesterol; *p*-value using ANOVA or chi-square test.

**Table 2 ijerph-19-01330-t002:** Participant characteristics according to metabolic syndrome and sex.

Variables	Males			Females		
	MetS (*n* = 1309)	Non-MetS (*n* = 2318)	*p*	MetS (*n* = 1899)	Non-MetS (*n* = 2837)	*p*
Age (y)	60.38 ± 0.26	60.94 ± 0.22	0.097	64.47 ± 0.28	60.43 ± 0.21	<0.0001
Sarcopenia, *n* (%)	331 (22.0)	350 (13.4)	<0.0001	527 (27.3)	424 (14.7)	<0.0001
BMI (kg/m^2^), mean (SD)	25.37 ± 0.10	22.93 ± 0.08	<0.0001	25.80 ± 0.10	23.23 ± 0.06	<0.0001
<18.5 (underweight), *n* (%)	9 (0.6)	119 (4.5)		12 (0.7)	109 (3.6)	
<25 (normal-weight), *n* (%)	581 (43.1)	1747 (74.1)		759 (40.9)	2070 (72.6)	
≥25 (overweight), *n* (%)	719 (56.3)	452 (21.4)		1128 (58.4)	658 (23.8)	
Smoking status, (%) (current-/ex-/non-smoker)	61.2/23.7/15.1	58.8/25.4/15.9	0.463	7.7/1.8/90.5	6.6/1.8/91.6	0.528
Drinking status (%) (current-/non-drinking)	71.6/28.4	69.7/30.3	0.344	23.0/77.0	30.2/69.8	<0.0001
Marital status, (%) (living with spouse)	90.9	92.7	0.085	60.4	71.9	<0.0001
Income (individual)						
Q1 (lowest) Q2 Q3 Q4 (highest)	24.6 25.0 23.8 26.6	24.1 26.3 24.7 24.9	0.733	26.6 26.6 24.3 22.6	23.6 23.7 27.2 25.5	0.014
Height (cm)	167.63 ± 0.212	166.79 ± 0.16	0.002	153.30 ± 0.18	153,98 ± 0.15	0.003
Weight (kg)	71.39 ± 0.35	63.89 ± 0.25	<0.0001	60.73 ± 0.26	55.14 ± 0.18	<0.0001
Fasting glucose (mg/dL)	116.13 ± 0.21	98.99 ± 0.55	<0.0001	111.38 ± 0.82	94.41 ± 0.37	<0.0001
Waist circumference (cm)	90.30 ± 0.28	82.55 ± 0.21	<0.0001	87.76 ± 0.26	78.89 ± 0.21	<0.0001
Triglyceride	230.53 ± 5.13	125.05 ± 3.16	<0.0001	184.88 ± 2.84	105.06 ± 1.29	<0.0001
Systolic BP (mmHg)	133.58 ± 0.55	123.01 ± 0.47	<0.0001	135.61 ± 0.46	121.48 ± 0.45	<0.0001
Diastolic BP (mmHg)	84.31 ± 0.38	78.43 ± 0.30	<0.0001	80.91 ± 0.29	75.83 ± 0.26	<0.0001
HDL-cholesterol	39.79 ± 0.33	48.36 ± 0.32	<0.0001	42.74 ± 0.25	52.37 ± 0.29	<0.0001
Total cholesterol	188.97 ± 1.34	186.30 ± 0.85	0.088	202.74 ± 1.17	199.65 ± 0.84	0.030
Skeletal muscle index	0.86 ± 0.00	0.90 ± 0.00	<0.0001	0.56 ± 0.00	0.60 ± 0.00	<0.0001

Data are presented as means ± SD or number (%). MetS; metabolic syndrome, BMI; body mass index, BP; blood pressure HDL-C; high density lipoprotein-cholesterol; *p*-value using ANOVA or chi-square test.

**Table 3 ijerph-19-01330-t003:** Metabolic syndrome components according to sarcopenia and sex.

Variables	Males		*p*	Females		*p*
	Sarcopenia	Non-Sarcopenia		Sarcopenia	Non-Sarcopenia	
^a^ High fasting glucose	55.3 ± 2.2	44.9 ± 1.1	<0.0001	46.2 ± 2.1	34.9 ± 0.9	<0.0001
^b^ Abdominal obesity	41.5 ± 2.4	27.5 ± 1.0	<0.0001	54.3 ± 2.2	34.4 ± 1.1	<0.0001
^c^ High triglyceride	47.3 ± 2.1	38.4 ± 1.1	<0.0001	40.3 ± 2.0	30.0 ± 1.0	<0.0001
^d^ High blood pressure	56.7 ± 2.5	48.4 ± 1.2	0.002	55.9 ± 2.1	44.9 ± 1.1	<0.0001
^e^ Low HDL-C	45.0 ± 2.5	36.9 ± 1.2	0.002	62.0 ± 1.9	59.3 ± 1.1	0.213
MetS	49.6 ± 2.3	35.0 ± 1.1	<0.0001	55.1 ± 2.0	35.9 ± 1.0	<0.0001

Data were presented as means ± SD or number (%). HDL-C; high density lipoprotein-cholesterol, MetS; metabolic syndrome; ^a^ High fasting glucose level is defined as FBG ≥100 mg/dL; ^b^ Abdominal obesity is defined as waist circumference >90 cm (male) or >85 cm (female); ^c^ High triglyceride level is defined as TG ≥150 mg/dL; ^d^ Low HDL-C level is defined as HDL-C <40 mg/dL (male) or <50 mg/dL (female); ^e^ High blood pressure is defined as SBP ≥130 mmHg or DBP ≥85 mmHg.

**Table 4 ijerph-19-01330-t004:** Odds ratios for sarcopenia according to metabolic syndrome stratified by sex.

	MetS	OR (95% CI)	*p*
Crude	Males	1.827 (1.496–2.231)	<0.0001
	Females	2.189 (1.818–2.635)	<0.0001
	Total	2.029 (1.766–2.331)	<0.0001
Model 1	Males	1.970 (1.597–2.430)	<0.0001
	Females	1.792 (1.489–2.157)	<0.0001
	Total	1.849 (1.608–2.126)	<0.0001
Model 2	Males	1.972 (1.599–2.432)	<0.0001
	Females	1.794 (1.489–2.160)	<0.0001
	Total	1.822 (1.586–2.095)	<0.0001
Model 3	Males	1.957 (1.587–2.413)	<0.0001
	Females	1.779 (1.478–2.141)	<0.0001
	Total	1.822 (1.586–2.095)	<0.0001

Model 1: Adjusted for age; Model 2: Model 1 + smoking and drinking status; Model 3: Model 2 + individual income and marital status; Reference category: non-sarcopenia.

## Data Availability

All data were anonymized and can be downloaded from the website at https://knhanes.kdca.go.kr/knhanes (accessed on 12 December 2021).

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
