# Peer review of "Sarcopenia Is Associated with Metabolic Syndrome in Korean Adults Aged over 50 Years: A Cross-Sectional Study"

_ijerph, 2022, doi:10.3390/ijerph19031330_

Round 1

Reviewer 1 Report

This study, based on data from the Korea National Health and Nutrition Survey data, is interesting. It seeks to demonstrate a correlation between sarcopenia and the metabolic syndrome. However, this study does not reveal whether there is a concordance between these two phenomena, or a real cause-and-effect relationship.

If there are 8,363 subjects in this study, only 681 of them have sarcopenia. This is ultimately quite low. On the other hand, the assessment of sarcopenia is not based on validated criteria, reference 19 does not correspond at all to this question. I wonder about the appropriateness of measuring the body mass index. Please provide bibliographic references. At this age, an estimate of fat mass, based on the measurement of skin folds, would not be more interesting, in connection with sarcopenic obesity in the elderly?

In the elderly, the stage of undernutrition is set at 21 rather than 18.5 as suggested in Table 1. What is the advantage of having the number or percentage of subjects with BMI < 18.5? Should we not rather know the percentage of undernourished subjects?

Line 154 : was evident or was evidenced ?

In the introduction, it is written : Various studies have identified a link between sarcopenia and MetS, other studies have found no correlation, and yet other studies have shown that the relationship is dependent on gender [15-18]. However, there are no references in the discussion that discuss this link, yet those references exist. The discussion should be enriched by this data, explaining why there is no consensus in the literature.

Table 4 : Is there not an error for this value 2.209?

Reviewer 2 Report

Dear authors,

thank you for this paper. It is clear in structure and contents. Methods, analysis and discussion are comprehensible and clear. Limitations are described  and discussed in an appropriate manner. The findings in the conclusion could be offered in a broader context, e.g. with reference to health.  A short view what can be done for public health according to the findings could be given.

Reviewer 3 Report

Major comment:

  1. It is unclear why the authors would like to study the association between metabolic syndrome and sarcopenia.
  2. How covariates were selected is not so clear. 
  3. The conclusion is also a bit strong with the word "independently", better to modify this. 

Minor comment:

  1. Table 4 could change the columns to males, females, and total, with OR(95%CI) combined as one sub-column and p-value as another sub-column. 
  2. The discussion section, "Table III" should be Table 3. 

Round 2

Reviewer 3 Report

Thanks for addressing the comments